# Characterizing and Mitigating Reasoning Drift in Large Language Models

**Yufeng Zhang**[1,2]**, Xuepeng Wang**[1,3*]**, Lingxiang Wu**[1,3]**, Jinqiao Wang**[1,2,3*]
[1]Institute of Automation, Chinese Academy of Sciences
[2]School of Artificial Intelligence, University of Chinese Academy of Sciences
[3]Wuhan AI Research
{yufeng.zhang, xuepeng.wang}@ia.ac.cn
{lingxiang.wu, jqwang}@nlpr.ia.ac.cn

## Abstract

While chain-of-thought prompting enables powerful multi-step reasoning in Large Language Models (LLMs), the stochastic nature of the generation process undermines its reliability. In this work, we first analyze thousands of reasoning paths to identify **Reasoning Drift**, a key failure mode where models get locked into flawed reasoning patterns. We reveal that the manifestation of drift is a complex interplay between universal functional tendencies and unique, model-specific signatures. Based on the diagnosis, we propose Reasoning-Aware Activation Steering, a novel inference-time intervention method to gently nudge the model's activations away from pathological patterns. We pre-compute a library of vectors from contrastive functional transitions and apply them dynamically. Experiments show that our method effectively mitigates the drift problem and boosts accuracy. Additionally, it generalizes to out-of-distribution tasks, demonstrating a deeper capture of valid reasoning principles.

## 1 Introduction

Recently, large language models (LLMs) have demonstrated consistently improving performance across a wide range of reasoning tasks (Havrilla et al., 2024; Plaat et al., 2024; Ke et al., 2025; Chen et al., 2025). By generating multi-step chains of thought (CoT) (Wei et al., 2022), LLMs can provide explicit traces of their reasoning process. However, the generation of these thought chains is inherently stochastic. A growing body of research (Wang et al., 2023; Lightman et al., 2024; Bigelow et al., 2025; Bogdan et al., 2025) has shown that such randomness can significantly affect the reliability of reasoning: simply replacing certain sentences, which represent discrete and human-readable thought steps, along the reasoning path can lead to entirely different outcomes.

To mitigate the negative impact of stochasticity, several strategies have been proposed, such as self-consistency (Wang et al., 2023; Wan et al., 2025; Taubenfeld et al., 2025) and path search (Yao et al., 2023; Xie et al., 2023; Leang et al., 2025). These methods typically rely on extensive sampling and then selecting the most promising reasoning paths or final answers. While these approaches have been shown to improve reasoning accuracy, they face two major limitations. First, they often incur substantial computational cost due to heavy dependence on large-scale sampling and state exploration without clear guidance. Second, and more importantly, they treat LLMs largely as black boxes. By merely increasing trials rather than probing the internal mechanisms of reasoning, these methods offer limited insight into how LLMs actually reason, or why they sometimes fail.

In this work, we aim to uncover and analyze the deeper mechanisms underlying LLM reasoning. We begin with a fundamental question: why does resampling individual steps lead to such drastic differences in outcomes? Leveraging the recently released Math-Rollout dataset (Bogdan et al., 2025), we systematically investigate how substituting different candidate sentences at intermediate steps influences the final success of reasoning, and further explore the underlying transition patterns that most strongly determine reliable outcomes.

---

*Corresponding authors.

Our analysis yields two primary discoveries. First, we identify a "funneling effect" in the reasoning process: it begins in a state of high plasticity where outcomes are volatile, but early-stage choices rapidly solidify the trajectory, drastically reducing the potential for later-stage correction and largely determining the final outcome. Second, we diagnose the underlying mechanism for this phenomenon, a failure mode we term **Reasoning Drift**. We find that drift is not random but is strongly correlated with specific, pathological shifts in the functional behavior between steps. For instance, models are prone to drift when they prematurely enter plan generation without adequate problem setup, or exhibit overconfidence by systematically avoiding corrective uncertainty management.

To mitigate the reasoning errors stemming from this drift, we propose a novel intervention inspired by activation steering (Turner et al., 2023; Rimsky et al., 2024; Arditi et al., 2024; Venhoff et al., 2025). Our method, termed Reasoning-Aware Activation Steering, is designed to be lightweight, interpretable, and targeted. By contrasting preferred and dispreferred functional shifts from the dataset, we extract a library of fine-grained steering vectors, each capable of gently nudging a deviating reasoning process back onto a productive path. During inference, we compute a real-time drift score to dynamically determine when and how to apply these vectors. Experiments demonstrate that our method is both effective in improving reasoning reliability and broadly applicable across tasks.

Our contributions are thus threefold:

- We conduct fine-grained analysis of the functional dynamics of reasoning trajectories. From this, we identify and characterize a critical failure mode we term *reasoning drift*, demonstrating that failures are systematically preceded by specific transitions.

- We propose a novel, reasoning-aware activation steering method. Our approach is lightweight, interpretable, and targeted. It detects the drift likelihood and applies precomputed steering vectors to gently guide the model's activations away from failure modes.

- Experimental validation shows that our method can mitigate the drift problem and improve reasoning accuracy. Furthermore, it is generalizable to out-of-distribution datasets.

## 2 ANALYSIS OF STEP-BY-STEP REASONING

Reasoning in LLMs is fundamentally a non-deterministic process. This inherent stochasticity, while a source of creativity, also undermines their reliability, as a model can generate both valid and fallacious reasoning paths for the very same problem (Wang et al., 2023; Lightman et al., 2024; Bigelow et al., 2025; Bogdan et al., 2025). Our investigation begins with a fundamental question: why merely resampling a single sentence in a reasoning chain can flip the final outcome from correct to incorrect, or vice versa. To quantify this phenomenon, we first establish an analytical framework.

### 2.1 PRELIMINARY DATASET

Probing the intricacies of model reasoning requires a fine-grained comparative analysis of successful versus failed trajectories. Such a framework is essential to pinpoint precisely where and how an LLM's thought process diverges under its inherent stochasticity. For this purpose, we leverage the recently proposed Math-Rollout dataset (Bogdan et al., 2025) as an analytical testbed. This dataset provides a curated collection of 40 correct and 40 incorrect solutions to mathematical problems, generated by the DeepSeek-R1-Distill-Llama-8B and DeepSeek-R1-Distill-Qwen-14B models. Each solution is decomposed into sentence-level reasoning steps.

Crucially, each reasoning step is accompanied by 100 stochastic rollouts (i.e. alternative paths the model could have taken) annotated with their final outcome. This provides a rich, counterfactual view into the model's decision space at each step. Furthermore, each step is annotated with one of eight functional categories to delineate its role in the problem-solving process: **Problem Setup**, **Plan Generation**, **Fact Retrieval**, **Active Computation**, **Result Consolidation**, **Uncertainty Management**, **Self Checking**, and **Final Answer Emission**. Details of the dataset are provided in Appendix C. While the dataset provides this powerful lens, the dynamics of why and how models navigate these steps successfully or fall into failure modes remain unexplored. This is the central investigation of our work.

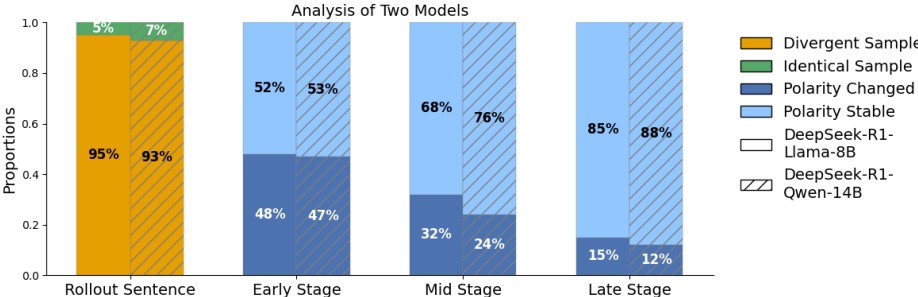

Figure 1: Analysis of rollout stability. The leftmost group shows the proportion of identical versus divergent samples after resampling from the original sentence. The three rightmost groups show the outcome polarity flip rate at the early, mid, and late stages of the reasoning process. For example, for DeepSeek-R1-Distill-Llama-8B, while only 5% of samples are identical to the original, 48% of resamples in the early stage lead to a flip in the final outcome's polarity.

## 2.2 PROBING REASONING INSTABILITY

We define the *Outcome Polarity Flip Rate* as the probability that a rollout's final outcome (success or failure) differs from the original reasoning path's outcome. We then segment the reasoning process into three distinct phases to analyze its temporal dynamics: the Early Stage (the first third of steps), the Mid Stage (the middle third), and the Late Stage (the final third).

As illustrated in Figure 1, our first observation is the inherent stochasticity of the models: rollouts are textually identical to the original sentence in only a mere 5%~7% of cases. For the vast majority ($\geq$93%) of rollouts that introduce textual variations, the impact on the final outcome is profound. The Outcome Polarity Flip Rate in the Early Stage is exceptionally high, exceeding 47% for both models. This rate diminishes to approximately 30% in the Mid Stage and continues to fall thereafter.

**This demonstrates a "funneling effect" in the reasoning process.** The initial steps are highly plastic and serve as critical inflection points, where even slight variations in expression can have a massive, almost coin-flip-like, impact on the final trajectory. As reasoning progresses, the path solidifies and becomes increasingly deterministic. While this resonates with the micro-level observations of Ji et al. (2025), our analysis highlights this phenomenon at a more systemic and global scale. This raises the crucial next question: Are these influential, early-stage variations simply random noise, or do they represent meaningful transitions in the model's underlying reasoning pattern?

## 2.3 FUNCTIONAL DYNAMICS OF TRANSITIONS

To answer the question posed above, we now shift our focus from textual variation to the underlying transitions in reasoning patterns. We seek to understand how the functional role of a rollout sentence (at step $t$) diverges from the role of the preceding step that prompted it (at step $t - 1$). Our goal is to determine how these specific one-step transitions correlate with the eventual success or failure of the reasoning chain.

To enable a large-scale, automated analysis of these dynamics, we trained a robust DistilBERT classifier (Sanh et al., 2019) on the annotated functional categories from the Math-Rollout dataset (see Appendix D for training details and performance). We then applied this classifier to annotate every rollout sentence with its corresponding functional label. This process transformed our raw text data into a structured sequence of steps, enabling us to construct a comprehensive transition matrix that quantifies the tendency for the model to move from one functional step to another.

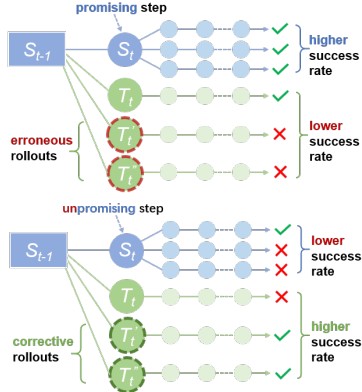

Figure 2: Illustration of different rollouts and steps.

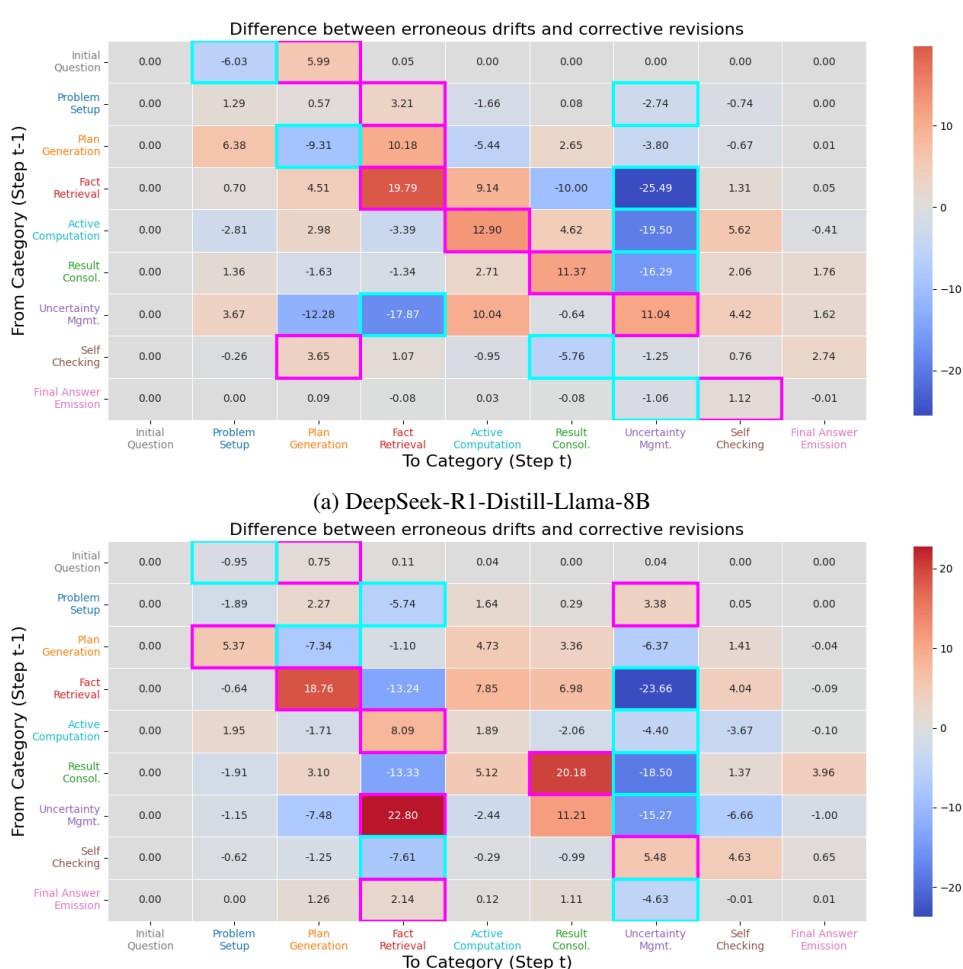

(a) DeepSeek-R1-Distill-Llama-8B

(b) DeepSeek-R1-Distill-Qwen14B

Figure 3: Heatmap of step transitions contrasting erroneous drifts and corrective revisions. The matrix is constructed by subtracting the frequency of corrective revisions ($S_{t-1} \to T_t \mid S_t = $ *unpromising*, $T_t = $ *corrective*) from that of erroneous drifts ($S_{t-1} \to T_t \mid S_t = $ *promising*, $T_t = $ *erroneous*). Red cells indicate transitions that are more frequent in failed reasoning paths, whereas blue cells highlight transitions more common in successful reasoning paths. The neon blue and purple outlines highlight, for each row, the minimum and maximum values, respectively.

Notably, to fully capture the entire reasoning trajectory, we expand the set of eight functional categories (from Section 2.1) to nine by introducing a special-purpose "zeroth" category: **Initial Question**. This serves as the universal starting point for every reasoning chain, enabling our transition analysis to model the critical move from the problem prompt to the opening line of reasoning.

We now introduce the diagnostic framework used to assess the health of each transition within it. Formally, a reasoning path consists of intermediate steps $S_1, S_2, ..., S_{t-1}, S_t, ..., S_M$ with final outcome $A_{S_t}$. As shown in Figure 2, a rollout from step $S_t$ means replacing $S_t$ with a candidate $T_t$ and completing the reasoning until the final answer $S_1, S_2, ..., S_{t-1}, T_t, ..., T_N, A'_{S_t}$. Each rollout is evaluated by whether the resulting outcome is successful or not:

- **Corrective Rollout:** A rollout sentence $T_t$ is corrective (green nodes with green borders in Figure 2) if the reasoning path it produces leads to a successful final outcome.

- **Erroneous Rollout:** A rollout sentence $T_t$ is erroneous (green nodes with red borders in Figure 2) if it leads to a failed final outcome.

Then, we define the potential of $S_t$ (blue nodes in Figure 2) by comparing the success rate of keeping $S_t$ versus replacing it with rollouts $T_t$:

- **Unpromising Step:** A step $S_t$ is unpromising if, on average, its rollouts achieve higher success rate than keeping $S_t$. In this case, resampling from $S_t$ is beneficial.

- **Promising Step:** Conversely, $S_t$ is promising if keeping it achieves higher success rate than resampling. In this case, resampling is detrimental.

This diagnostic lens allows us to isolate the most telling events in a reasoning trajectory. Our analysis will concentrate on two opposing dynamics: a) the "**the correction scenario**," where we study the state transition from a preceding step $S_{t-1}$ to a *corrective rollout* $T_t$ that remedies an *unpromising step* $S_t$; and b) the "**the drift scenario**," where we study the transitions from $S_{t-1}$ that a *promising step* $S_t$ derails into an *erroneous rollout* $T_t$.

### 2.4 REASONING DRIFT PHENOMENON

Figure 3 visualizes the heatmap of our difference matrix (*Drift Matrix - Correction Matrix*), for which the individual matrices are provided in Appendix E. Here, hotter colors signify pathological transitions strongly associated with failure, and cooler colors denote salutary transitions characteristic of correction. This diagnostic map reveals several critical insights into various failure modes, which we collectively refer to as **Reasoning Drift**.

First, at the very beginning of the process, we observe a shared failure archetype. The transition from the *initial question* to *plan generation* is a strong indicator of an erroneous path, suggesting that models often **fail by rushing to a solution** without sufficient *problem setup*. Furthermore, we find a universal characteristic of failure is the **avoidance of the *uncertainty management* step**. The strong negative values for all transitions into this step imply that entering a step of self-reflection is a powerful corrective mechanism, one that failing trajectories systematically bypass in a display of erroneous overconfidence.

Most strikingly, the heatmap reveals **unique, model-specific drift signatures**. Llama's drift is predominantly inertial: it manifests as a strong tendency to get stuck in unproductive self-loops, as seen in the strong positive values along the matrix diagonal. We term this **Inertial Drift**. Qwen, conversely, exhibits a more chaotic form of drift. Its failure is not characterized by getting stuck, but by making erratic, non-linear jumps, often backtracking to previous functional states. This pattern, visible in the hotter lower triangle of its matrix, we classify as **Chaotic Drift**.

In summary, our analysis reveals that *reasoning drift* is not a monolithic phenomenon but manifests in model-specific signatures. Having diagnosed these distinct pathological dynamics, the critical question becomes: can we develop a targeted intervention to actively steer the model away from these failure modes and guide it back towards a productive reasoning path?

## 3 PROPOSED METHOD: REASONING-AWARE ACTIVATION STEERING

To address the challenge outlined above, we propose a novel, inference-time intervention method: Reasoning-Aware Activation Steering (RAAS). Prior work has demonstrated the efficacy of activation steering across various domains (Subramani et al., 2022; Turner et al., 2023; Rimsky et al., 2024; Han et al., 2024; Arditi et al., 2024), yet its application to the nuanced, multi-step nature of reasoning remains underexplored. Building upon our fine-grained analysis of reasoning behaviors, we adapt this technique to intervene in the subtle and complex transitions that determine reasoning outcomes.

Our method is designed to act as a corrective nudge in the model's activation space, precisely when it begins to exhibit the transition patterns characteristic of a potential drift. Unlike fine-tuning approaches that require extensive retraining, or inference-time methods like self-consistency and path search that rely on massive sampling, our method offers a lightweight, interpretable, and targeted way to enhance reasoning robustness.

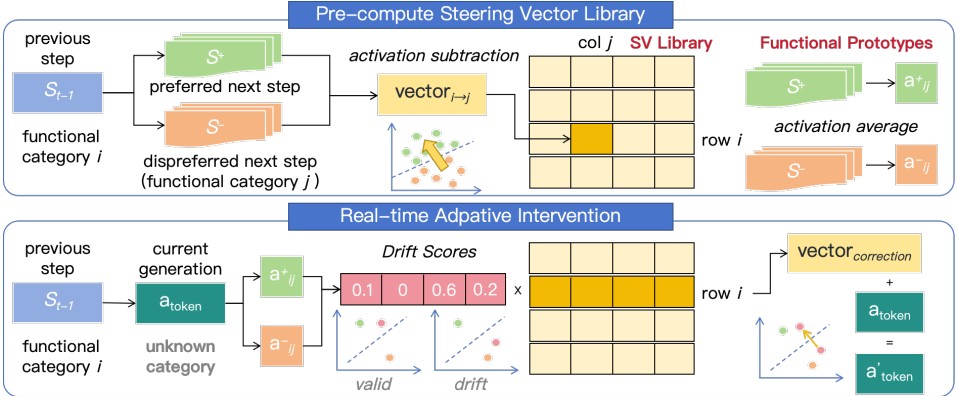

Figure 4: The overall architecture of reasoning-aware activation steering.

## 3.1 Constructing Contrastive Pairs

Our method is based on learning a directional vector from contrastive pairs. Formally, we construct a dataset $D$ where each entry represents a single reasoning choice, containing a shared context $S_{<t}$, a preferred next step $S^+$, and a dispreferred next step $S^-$. The assignment of these roles is determined by the two critical scenarios identified in our analysis (Section 2.3):

- **The Correction Scenario:** This occurs when the original step $S_t$ is unpromising. In this case, our goal is to steer the model towards a better path. We therefore assign the *corrective rollout $T_t$* as the preferred next step ($S^+$), and the original *unpromising step $S_t$* itself as the dispreferred next step ($S^-$).
- **The Drift Scenario:** This occurs when the original step $S_t$ is promising. Here, our goal is to steer the model away from the erroneous path it drifted into. We therefore assign the *promising step $S_t$* as the preferred next step ($S^+$), and the *erroneous rollout $T_t$* as the dispreferred next step ($S^-$).

Consequently, every entry in our final dataset $D$ is a triplet of the form $(S_{<t}, S^+, S^-)$. This structure allows us to systematically learn the vector differences between desirable and undesirable reasoning paths, always pointing from a less-preferred step to a more-preferred one.

## 3.2 Extracting the Behavioral Steering Vectors

Next, we proceed to pre-compute a comprehensive library of fine-grained steering vectors, each tailored to counteract a specific type of pathological functional transition. This library can be conceptualized as a matrix $V$ of vectors, where an element $v_{i \to j}$ represents the corrective vector for a transition from a step of category $i$ to an undesirable step of category $j$. It is crucial to clarify that the steering vector $v_{i \to j}$ is not designed to prevent all transitions from category $i$ to $j$, as many such transitions are valid and essential for correct reasoning. Instead, we aim to correct only the specific instances of that transition that are showing signs of drifting.

We then iterate through all possible transition pairs $(i, j)$ found in our dataset, where $i$ is one of the 9 source categories (defined in Section 2.1 and 2.3) and $j$ is one of the 8 possible destination categories (excluding *initial question*). For each pair, we create a corresponding subset of data, $D_{i \to j}$, which contains all triplets where $S_{t-1}$ has the label $i$ and $S^-$ has the label $j$.

The steering vector $v_{i \to j}$ is then computed by averaging the difference between the preferred and dispreferred activations within this specific subset:

$$v_{i \to j} = \mathbb{E}_{(S^+, S^-) \in D_{i \to j}}[\text{act}(S^+) - \text{act}(S^-)] \qquad (1)$$

where $\text{act}(S)$ is the activation representation of a sentence $S$. We define this as the mean-pooled vector of its token activations, $\mathbf{a}_L$, from the $L$-th transformer layer:

$$\text{act}(S) = \frac{1}{|S|} \sum_{k=1}^{|S|} \mathbf{a}_L(\text{token}_k) \tag{2}$$

This systematic process yields a library of up to $9 \times 8 = 72$ unique steering vectors. Each vector, therefore, acts as a gentle nudge to guide a potentially pathological instance of a transition back towards its salutary counterpart. Details of intervention layer selection are in Appendix F.

## 3.3 REAL-TIME DRIFT DETECTION

Armed with our library of fine-grained steering vectors, we face the challenge of applying them during sequential generation. While we can classify the preceding step $S_{t-1}$, the functional category of the current, partially-generated step $S_t$ is unknown. Backtracking or lookahead mechanisms are computationally prohibitive.

To overcome this, we propose a **Real-time Drift Score**, a real-time metric that predicts the model's instantaneous tendency to drift towards a specific pathological function. To compute this score, we first pre-calculate a set of **Functional Prototypes** from our dataset $D$. The pathological prototype $(\mathbf{a}_{i,j}^-)$ is the average activation of all failing examples, while the salutary prototype $(\mathbf{a}_{i,j}^+)$ is the average of all successful ones:

$$\mathbf{a}_{i,j}^- = \mathbb{E}_{(\cdot, S^-) \in D_{i \to j}}[\text{act}(S^-)], \mathbf{a}_{i,j}^+ = \mathbb{E}_{(S^+, \cdot) \in D_{i \to j}}[\text{act}(S^+)] \tag{3}$$

During inference, as the model generates each token for the current step $S_t$, we extract its activation as $\mathbf{a}_{\text{token}}$. The real-time drift score is then formulated to be non-zero only when the token's activation is demonstrably closer to a pathological prototype than its salutary counterpart. Formally, the score for each potential destination category $j$ is:

$$\text{DriftScore}(i, j) = \mathbb{I}\left(\text{cosine}(\mathbf{a}_{\text{token}}, \mathbf{a}_{i,j}^-) > \text{cosine}(\mathbf{a}_{\text{token}}, \mathbf{a}_{i,j}^+)\right) \cdot \text{cosine}(\mathbf{a}_{\text{token}}, \mathbf{a}_{i,j}^-) \tag{4}$$

where $\mathbb{I}(\cdot)$ is the indicator function, which returns 1 if the condition is met and 0 otherwise. The score is therefore zero if the token is closer to the "good" prototype. If it is closer to the "bad" prototype, the score's magnitude is equal to its cosine similarity to that pathological prototype $(\mathbf{a}_{i,j}^-)$. A higher $\text{DriftScore}(i, j)$ thus provides a stronger signal that the current generation is not only directionally incorrect but also strongly aligned with a known failure mode $j$.

## 3.4 INFERENCE TIME ADAPTIVE STEERING

Finally, we integrate our pre-computed components (the steering vector library and the functional prototypes) into a cohesive online algorithm for inference-time steering. This algorithm provides continuous, fine-grained course correction throughout the generation process.

The process is executed at every token generation step (Wang et al., 2024). First, we take the completed previous step, $S_{t-1}$ (or the *initial question* prompt if $t = 1$), and apply our pre-trained classifier (Section 2.3) to determine its functional category, $i$. Then, using the current token's activation, we compute the $\text{DriftScore}(i, j)$ for all 8 potential destination categories $j$, as defined in Section 3.3. This score quantifies the model's instantaneous tendency to drift towards each pathological category. Next, we compute a dynamic, context-aware correction vector, $v_{\text{correction}}$, by taking a weighted sum of all steering vectors originating from the current context $i$. The weights for this sum are the real-time drift scores themselves:

$$v_{\text{correction}} = \sum_{j=1}^{8} \text{DriftScore}(i, j) \cdot v_{i \to j} \tag{5}$$

This aggregated correction vector is added to the original token activation to guide the generation:

$$\mathbf{a}'_{\text{token}} = \mathbf{a}_{\text{token}} + v_{\text{correction}} \tag{6}$$

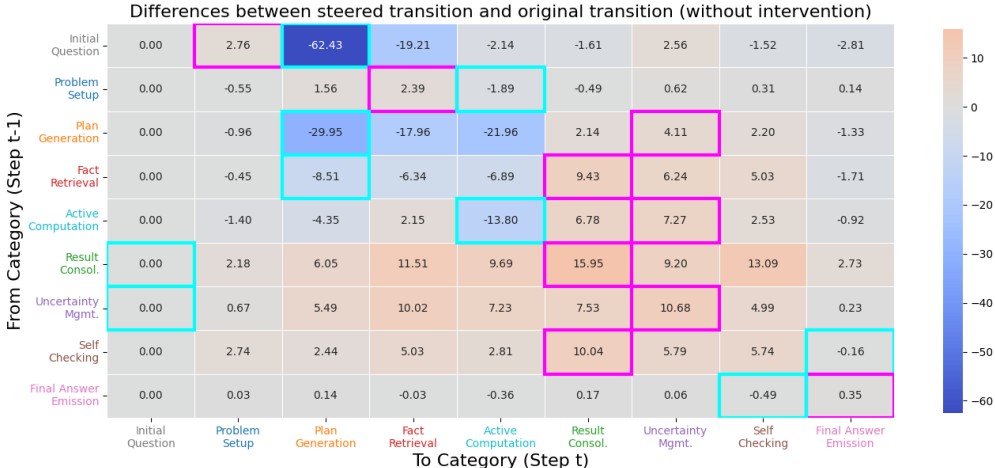

Figure 5: Heatmap of the difference matrix, calculated by subtracting the original transition matrix (without intervention) from the transition matrix of our steered model on R1-Distill-Llama-8B on GSM8K. Positive values (hotter colors indicate transitions that our method promotes, while negative values (cooler colors) signify transitions it suppresses.

# 4 EXPERIMENTS

In this section, we conduct a series of experiments to empirically validate the effectiveness of our proposed online steering method in improving reasoning capabilities.

## 4.1 EXPERIMENTAL SETUP

For our primary experiments, we employ the same base model used for vector extraction, R1-Distill-Llama-8B and R1-Distill-Qwen-14B. To assess the method's performance, we evaluate it on a suite of challenging mathematical reasoning benchmarks similar in nature to the Math-Rollout dataset: GSM8K (Cobbe et al., 2021), AIME2024 (MAA, 2024), AIME2025 (OpenCompass, 2025), and GPQA-Diamond (Rein et al., 2024). Details of implemention are in Appendix G. Our evaluation is structured around the following key research questions (RQs):

- **RQ1 (Mechanistic Validation):** How effectively does our method mitigate the specific patterns of the drift problem identified in our analysis?
- **RQ2 (Performance Impact):** Does the mitigation of reasoning drift translate into tangible improvements in final task accuracy?
- **RQ3 (Generalization):** Is our proposed framework for extracting and applying steering vectors a general approach applicable to other base models?

## 4.2 MITIGATION OF DRIFT (RQ1)

To answer RQ1, we first investigate the mechanistic impact of our steering method on the model's reasoning process. Figure 5 presents a heatmap of the difference matrix, calculated by subtracting the original transition matrix (without intervention) from the transition matrix of our steered model on Llama. Results for Qwen are in Appendix H. For each problem, we normalize the transitions by type according to their total number within that problem. Positive values (hotter colors) indicate transitions that our method promotes, while negative values (cooler colors) signify transitions it suppresses.

The results provide a clear affirmative answer. At the very start of the process, our method acts as a crucial guardrail, discouraging the model from prematurely jumping from the *initial question* into *plan generation* and instead promoting a more deliberate *problem setup*. Furthermore, the intervention consistently boosts the likelihood of entering the *uncertainty management* step from all

Table 1: Comparison results (%) across different reasoning tasks. Methods with $^*$ denote the results are copied from Leang et al. (2025), while those without are reproduced.

| Models | Methods | GSM8K | AIME2024 | AIME2025 | GPQA-Diamond |
|---|---|---|---|---|---|
| R1-Distill-Llama-8B | Vanilla$^*$ | 73.67($\pm$0.32) | 37.96($\pm$1.52) | 29.63($\pm$0.37) | 42.87($\pm$1.07) |
| | SC$^*$ | 74.01($\pm$0.70) | 38.89($\pm$1.67) | 25.00($\pm$0.37) | 42.10($\pm$1.77) |
| | PiCSAR$^*$ | 76.42($\pm$0.16) | 47.78($\pm$4.01) | 33.33($\pm$1.11) | 47.31($\pm$0.17) |
| | Vanilla | 82.45($\pm$0.78) | 34.99($\pm$1.67) | 25.56($\pm$4.44) | 50.50($\pm$1.51) |
| | CAA$_{um}$ | 85.06($\pm$0.36) | 45.00($\pm$3.49) | 30.00($\pm$2.35) | 52.52($\pm$1.16) |
| | Ours | 87.56($\pm$0.30) | 55.56($\pm$1.49) | 31.70($\pm$1.66) | 55.52($\pm$0.90) |
| R1-Distill-Qwen-14B | Vanilla | 93.69($\pm$0.98) | 54.44($\pm$2.23) | 26.67($\pm$3.33) | 55.55($\pm$1.01) |
| | CAA$_{um}$ | 95.36($\pm$0.38) | 53.33($\pm$2.35) | 28.89($\pm$2.33) | 52.02($\pm$1.63) |
| | Ours | 95.60($\pm$0.42) | 62.49($\pm$1.66) | 36.67($\pm$1.92) | 57.57($\pm$0.76) |

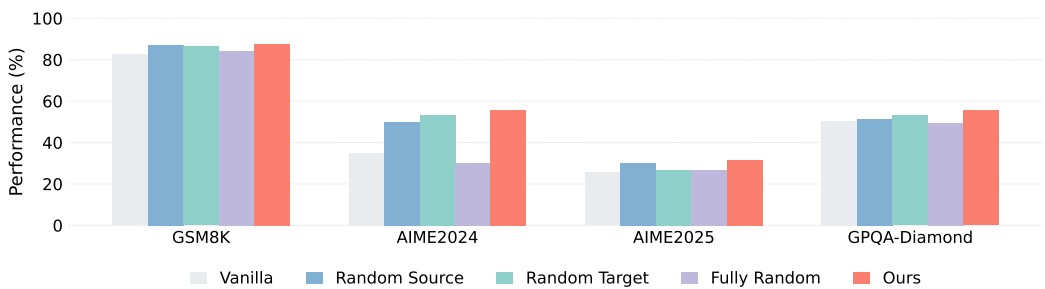

Figure 6: Ablation study of steering variants. While partially mismatched vectors (Random Source/Target) offer marginal regularization benefits over the vanilla baseline, fully random interventions severely disrupt reasoning, highlighting the necessity of functional context.

other steps. This demonstrates a significant reduction in the model's overconfidence, forcing it to adopt a more reflective and cautious reasoning style, a benefit that echoes the findings of Bogdan et al. (2025). The heatmap also provides direct evidence of model-specific drift mitigation. While strong inertial patterns persist in the late stages of reasoning (the bottom-right of the matrix remains hot), we see a marked reduction in the pathological self-transition loops in the early-to-mid stages (cooler diagonals in the top-left and middle). This demonstrates that our steering mechanism is most effective during the early, high-plasticity phase.

## 4.3 RESULT COMPARISON (RQ2)

To answer RQ2, we evaluate our approach against three representative baselines: vanilla CoT, Self-Consistency (SC) (Wang et al., 2023), and the PiCSAR (Leang et al., 2025) path-selection method. Additionally, we conducted an experiment where we exclusively selected the single vector (Rimsky et al., 2024) associated with *uncertainty management* to investigate whether it could mitigate the model drift caused by insufficient self-reflection, termed CAA$_{um}$.

Table 1 summarizes the accuracy of all methods across several challenging mathematical reasoning datasets analogous to Math-Rollout. Overall, the results show that our method consistently improves the reasoning accuracy. The performance gains are particularly pronounced on highly complex problem sets like AIME 2024 and AIME 2025, which demand sophisticated, multi-step logical deduction. Notably, these evaluation datasets are out-of-distribution (OOD) within the math domain with respect to the data used to extract the steering vectors. They demonstrate that our fine-grained functional vectors are not merely overfitting to the source distribution but have captured generalizable, fundamental principles of logical reasoning.

To further validate whether our method's success stems from generic activation perturbations, we conduct a targeted ablation study. Specifically, when the model is in step $i$ and exhibits drift towards $j$ (where the optimal intervention is $\mathbf{v}_{i \rightarrow j}$), we intentionally mismatch the contextual components. We introduce three randomized steering variants: 1) Random Source ($i_{\text{rand}} \rightarrow j$) that applies a

vector from a randomly sampled source category $i_{\text{rand}}$; 2) Random Target ($i \to j_{\text{rand}}$) that applies a vector from a randomly sampled target category $j_{\text{rand}}$; 3) Fully Random ($i_{\text{rand}} \to j_{\text{rand}}$) that applies a randomly selected vector from the library.

Figure 6 compares the three randomized variants against baselines. Overall, our proposed steering consistently achieves the highest reasoning accuracy. Interestingly, the Random Source and Random Target variants yield marginal improvements over the vanilla baseline. We infer that since all vectors are derived from the difference between preferred and dispreferred activations, they inherently encode a general corrective direction. Applying them acts as a beneficial test-time regularization. Conversely, the Fully Random variant severely degrades performance. These results confirm that while a generic corrective nudge offers slight benefits, precise context mapping ($i \to j$) is essential for maximizing reasoning improvements.

### 4.4 APPLICATION TO OTHER MODELS (RQ3)

To answer RQ3, we directly reuse the vectors learned on the distilled models and apply them to the corresponding non-distilled base models. We are interested in what happens under this zero-retraining transfer scenario. Table 2 presents the performance of our method when applied to Llama 3.1-8B and Qwen 2.5-14B, prior to their task-specific distilla-

Table 2: Results on other models.

| Model | Method | GSM8K | GPQA-D |
|---|---|---|---|
| Llama3.1-8B | Vanilla | 83.91 | 28.28 |
| | Ours | 85.22 | 29.10 |
| Qwen2.5-14B | Vanilla | 83.77 | 34.27 |
| | Ours | 84.46 | 35.18 |

tion. While the performance improvements are smaller compared to our primary model, the method is consistently applicable and produces positive effects across all tested models. These results indicate that our framework is generally applicable, though the magnitude of improvement depends on model-specific characteristics such as reasoning capacity and pretraining data.

## 5 RELATED WORK

**Chain-of-Thought and Advanced Reasoning Models.** Chain-of-Thought (CoT) prompting has become the de facto standard for eliciting multi-step reasoning in Large Language Models (LLMs) (Wei et al., 2022). Building upon this, a new generation of models exhibiting advanced "thinking" capabilities, such as DeepSeek-R1 (Guo et al., 2025), OpenAI's o3 (o3, 2025), and Anthropic's Claude series (Anthropic, 2025), have emerged. The models central to our analysis, R1-distill-llama-8B and R1-distill-qwen-14B, are direct distillations from the powerful DeepSeek-R1, inheriting its strong reasoning faculties and making them ideal subjects for our study.

**Addressing Stochasticity in Reasoning.** Despite these advances, two dominant strategies exist: sampling-based methods like self-consistency (Wang et al., 2023; Wan et al., 2025; Taubenfeld et al., 2025), which rely on majority voting, and search-based methods like Tree-of-Thoughts (Yao et al., 2023), beam search (Xie et al., 2023), and PiCSAR (Leang et al., 2025), which explore and prune multiple reasoning paths. Our work builds upon prior analyses of thinking models (Venhoff et al., 2025; Bogdan et al., 2025).

**Activation Steering for Process Intervention.** We draw inspiration from activation steering (Turner et al., 2023; Rimsky et al., 2024; Arditi et al., 2024; Venhoff et al., 2025), a technique that modifies model behavior by adding a steering vector to its internal activations during inference. This allows us to guide the model away from undesirable states and towards desirable ones, enhancing reliability without the significant computational overhead of massive resampling.

## 6 CONCLUSION

This paper tackled the challenge of stochasticity in LLM reasoning through a diagnostic lens. By analyzing the functional dynamics of reasoning, we identified and characterized Reasoning Drift as a primary failure mechanism. Our findings reveal that models drift into failure via distinct signatures. To combat these identified failures, we developed Reasoning-Adaptive Activation Steering, a targeted, inference-time method that demonstrates the effectiveness of process-oriented intervention. Our results show that this process-oriented approach not only enhances reasoning accuracy but also generalizes to different scenarios.

ACKNOWLEDGMENTS

This work was supported by the National Key R&D Program of China under Grant No.2024YFC3015501, National Natural Science Foundation of China (No. 62276260). We sincerely thank all reviewers and ACs for their insightful comments, time and efforts.

REPRODUCIBILITY STATEMENT

Our experiments rely on publicly available datasets (Math-Rollout, GSM8K, AIME benchmarks, and GPQA) and open-source models (Llama and Qwen series), all accessible via the Hugging Face Hub. All code, including data preprocessing, classifier training, vector extraction, and evaluation scripts, will be publicly released. Experiments were run with fixed random seeds, and key hyperparameters and environment specifications are detailed in the Appendix.

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

## A  LLMS USAGE STATEMENT

Regarding the writing process, our use of LLMs was strictly limited to refining and polishing the manuscript. The core ideas and overall structure of the paper were developed entirely by the authors. The LLM was employed solely for grammatical and syntactical checks, and all revisions were subsequently reviewed by the authors to ensure accuracy and fidelity to our original intent. As for the experiments, all code was manually written and executed. The only involvement of LLMs in our research was as a subject of the study itself. They were not used in any other part of the experimental process.

Table 3: Statistics of the math-rollout dataset.

| Models | R1-Distill-Llama-8B | R1-Distill-Qwen-14B |
|---|---|---|
| **# of Correct Base Solution** | 20 | 20 |
| **# of Avg. Steps in Correct** | 200.1 | 147.6 |
| **# of Total Rollouts in Correct** | 288500 | 184800 |
| **# of Incorrect Base Solution** | 20 | 20 |
| **# of Avg. Steps in Incorrect** | 174.3 | 176.1 |
| **# of Total Rollouts in Incorrect** | 281500 | 273600 |

## B   LIMITATIONS

We acknowledge the following limitations of our current work, which also point to possible directions for future research:

First, our methodology models reasoning as a first-order process, focusing on the transition from $S_{t-1}$ to $S_t$. This has proven effective for correcting local drifts but may not capture failures caused by longer-range, higher-order sequences of behaviors. Investigating these higher-order relationships would, however, lead to a combinatorial explosion in complexity and demand datasets with an even greater scale of rollouts to achieve statistical significance. Exploring more sophisticated sequential models to capture these long-range dynamics remains a direction for future work.

Second, our findings are based on the Llama and Qwen model families. While we uncover distinct, model-specific drift patterns, it is an open question how these signatures generalize to other prominent architectures, especially closed-source models whose internal states are not accessible for study. We believe our study provides a starting point and a methodological blueprint for a broader investigation into the common and unique reasoning behaviors across the entire LLM ecosystem.

## C   DETAILS OF DATASET

We adopt the taxonomy introduced by Bogdan et al. (2025), which classifies each sentence in a reasoning trace into eight categories:

- **Problem Setup:** Understanding and interpreting the original problem

- **Plan Generation:** Devising a reasoning plan based on the problem

- **Fact Retrieval:** Incorporating relevant background knowledge to support reasoning

- **Active Computation:** Carrying out explicit calculations or logical operations

- **Result Consolidation:** Integrating intermediate results or summarizing progress

- **Uncertainty Management:** Expressing confusion, re-evaluating intermediate steps, or backtracking when necessary

- **Self Checking:** Verifying the correctness of previous steps

- **Final Answer Emission:** Explicitly providing the final solution

This taxonomy provides a structured abstraction of sentence-level reasoning behaviors, allowing us to capture the fine-grained dynamics of reasoning traces. By distinguishing the functional role of each sentence, it offers a principled basis for analyzing, monitoring, and ultimately steering the reasoning process of LLMs. We build on this taxonomy to design mechanisms that address reasoning drift through dynamic steering.

Detailed statistics of the dataset are provided in Table 3.

Table 4: Classification report for our trained DitillBERT classifier.

|  | **Precision** | **Recall** | **F1-Score** | **Support** |
|---|---|---|---|---|
| Problem Setup | 0.89 | 0.67 | 0.76 | 12 |
| Plan Generation | 0.75 | 0.78 | 0.77 | 74 |
| Fact Retrieval | 0.91 | 0.88 | 0.89 | 207 |
| Active Computation | 0.92 | 0.91 | 0.91 | 243 |
| Result Consolidation | 0.75 | 0.74 | 0.75 | 82 |
| Uncertainty Management | 0.88 | 0.96 | 0.92 | 93 |
| Self Checking | 0.74 | 0.77 | 0.75 | 30 |
| Final Answer Emission | 0.88 | 1.00 | 0.93 | 7 |
| Macro Avg | 0.84 | 0.84 | 0.84 | 748 |
| Weighted Avg | 0.87 | 0.87 | 0.87 | 748 |

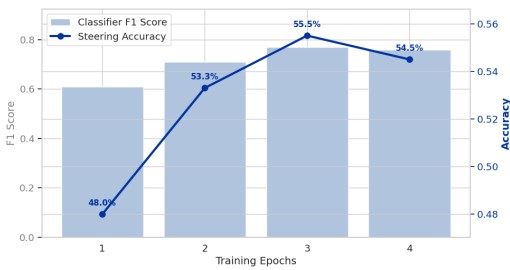

Figure 7: Impact of different classifier training degrees on the downstream activation steering accuracy. We select the AIME 2024 dataset as the experimental subject.

## D DETAILS OF TRAINED CLASSIFIER

To efficiently obtain the functional category for a large number of rollouts, we trained a DistilBERT[1] classifier on the pre-labeled sentences from the math-rollout dataset, chosen for its efficiency and reliability. Specifically, we used a total of 7,481 labeled samples, which were split into a 9:1 ratio for training and validation. The model was trained for 3 epochs with a learning rate of 5e-5 and weight decay of 0.01. The performance of the trained classifier is shown in Table 4.

To investigate how classifier quality influences final performance, we compared classifiers trained to varying degrees of accuracy. As shown in Figure 7, the accuracy of downstream activation steering is directly influenced by the classifier's training duration. We observed a clear trend: classifiers trained for fewer epochs performed worse on both category identification and the downstream task. However, performance degradation due to potential overfitting was observed after three epochs of training.

## E CORRECTION MATRIX & DRIFT MATRIX

Figures 8 and 9 display the original step transition matrices for the Llama and Qwen models, respectively. The corrective revisions matrix is constructed based on $(S_{t-1} \rightarrow T_t \mid S_t = unpromising, T_t = corrective)$, while the erroneous drift matrix is based on $(S_{t-1} \rightarrow T_t \mid S_t = promising, T_t = erroneous)$. To analyze the occurrence of drift, we subtracted the *Corrective Matrix* from the *Drift Matrix* to obtain the difference, as shown in Figure 3.

---

[1] https://huggingface.co/distilbert/distilbert-base-uncased

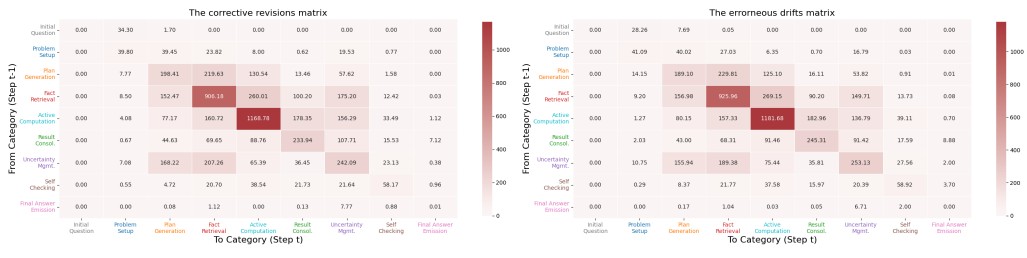

(a) R1-Distill-Llama-8B Corrective Matrix    (b) R1-Distill-Llama-8B Drift Matrix

Figure 8: Heatmap of step transitions matrix on R1-Distill-Llama-8B. The corrective revisions matrix is constructed by $(S_{t-1} \rightarrow T_t \mid S_t = unpromising, T_t = corrective)$, while the erroneous matrix is constructed by $(S_{t-1} \rightarrow T_t \mid S_t = promising, T_t = erroneous)$.

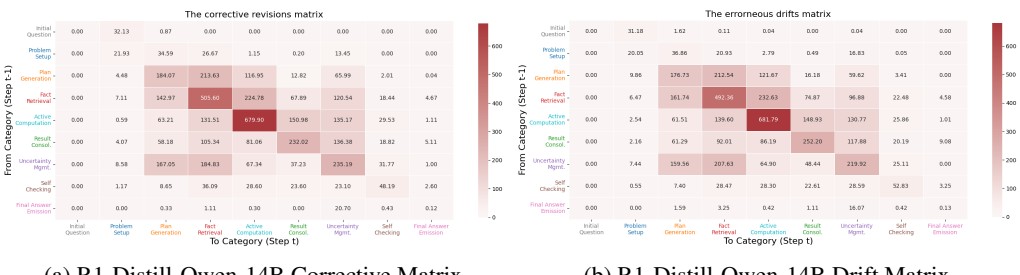

(a) R1-Distill-Qwen-14B Corrective Matrix    (b) R1-Distill-Qwen-14B Drift Matrix

Figure 9: Heatmap of step transitions matrix on R1-Distill-Qwen-14B.

## F  INTERVENTION LAYER SELECTION

The choice of which layer to apply the steering vector intervention to is a critical question, as intervening at layers that are too shallow or too deep may be suboptimal (Rimsky et al., 2024). Inspired by the methodology of *activation patching* (Syed et al., 2023; Nanda, 2023; Venhoff et al., 2025), we heuristically select the most impactful model layer for intervention. The core principle of this method is to measure the magnitude of change in activations, such as the KL divergence, after replacing a sentence with a counterfactual counterpart. Formally, the patching effect is measured by:

$$\Delta L = L \left( \mathbf{x}_{\text{clean}} \mid \text{do}(\mathbf{a} = \mathbf{a}_{\text{patch}}) \right) - L(\mathbf{x}_{\text{clean}}) \tag{7}$$

where $\mathbf{a}$ is the original activation, and $\mathbf{a}_{\text{patch}}$ is the counterfactual activation. We applied this approach across all functional categories to identify a single, comprehensively effective layer for intervention, as illustrated in Figure 10.

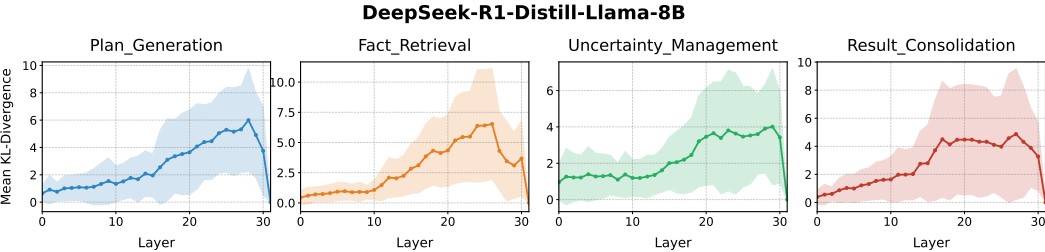

Figure 10: The patching effect across different model layers. The value measures how effective it is by applying the corresponding steering vector at each layer.

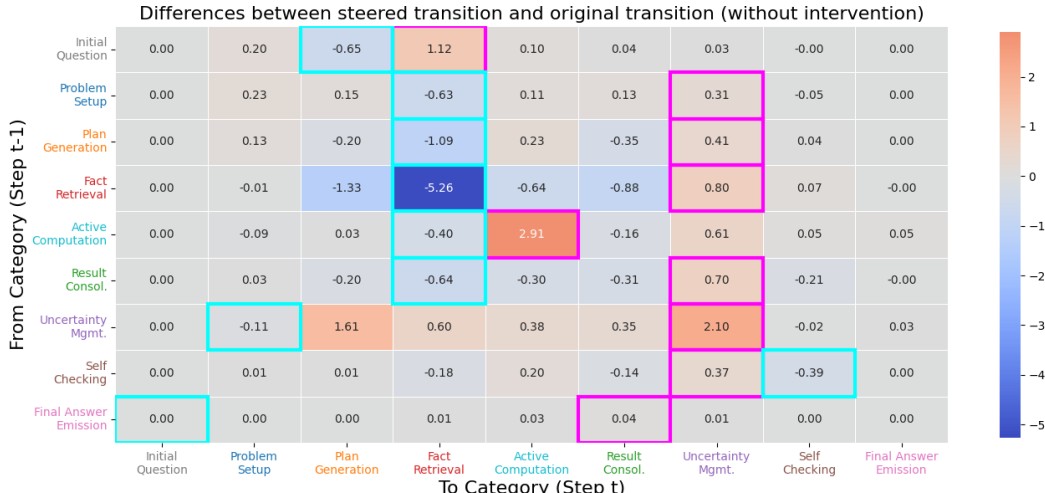

Figure 11: Heatmap of the difference matrix, calculated by subtracting the original transition matrix (without intervention) from the transition matrix of our steered model on R1-Distill-Qwen-14B on GPQA-Diamond.

## G  DETAILS OF IMPLEMENTATION

For our experiments, we selected the R1-distill-llama-8B[2] and R1-distill-qwen-14B models[3], which are the same models used for the steering vector extraction. We configured the hyperparameters according to the officially recommended settings, with a temperature of 0.6 and a top_p of 0.95. Instead of using the vLLM framework, we utilized the original model implementation to facilitate the acquisition of activations during the process. All experiments were conducted on four NVIDIA 4090 GPUs.

## H  DRIFT MITIGATION ON QWEN

Figure 11 shows the heatmap of the difference matrix calculated by subtracting the original transition matrix (without intervention) from the transition matrix of our steered model on Qwen. Positive values indicate transitions that our method promotes, while negative values signify transitions it suppresses.

We make the following observations:

- The transition from *initial question* to *plan generation* occurs less frequently. Instead, the model shows a preference for approaching the question through *fact retrieval*.

- There is an increased frequency of transitions leading to the *uncertainty management* state, which indicates the model exhibits more uncertainty during its process. This prompts the model to verify its steps more carefully before reaching a conclusion.

- The lower triangle of the matrix displays lower values (indicated by cooler colors), implying that the reasoning process becomes more linear and contains fewer non-sequential, backtracking jumps.

The findings, together with those of Llama, show that our method can effectively mitigate the drift problem.

## I  EVALUATION ON NON-MATH REASONING TASKS

---

[2]https://huggingface.co/deepseek-ai/DeepSeek-R1-Distill-Llama-8B
[3]https://huggingface.co/deepseek-ai/DeepSeek-R1-Distill-Qwen-14B

In our primary evaluations, we focused on mathematical reasoning benchmarks. Because our steering vectors were derived from the Math-Rollout dataset, these benchmarks served as rigorous out-of-distribution (OOD) tests within the same overarching mathematical domain. However, a fundamental question remains: do the pathological state transitions and corresponding steering vectors we identified capture domain-agnostic principles of logical reasoning, or are they overfitted to mathematical problem-solving?

Table 5: Results on non-math datasets.

| Model | Method | ReClor | LogiQA |
|---|---|---|---|
| R1-Distill-Llama-8B | Vanilla | 46.39 | 58.80 |
| | Ours | 48.85 | 61.00 |
| R1-Distill-Qwen-14B | Vanilla | 65.97 | 78.80 |
| | Ours | 66.51 | 81.40 |

To investigate the cross-domain applicability of our method, we extend our evaluation to broader, non-mathematical reasoning tasks. Specifically, we select ReClor (Yu et al., 2020) and LogiQA (Liu et al., 2020), two widely adopted benchmarks designed to assess reading comprehension and logical inference. Unlike mathematical datasets, these tasks require the model to parse complex textual premises and deduce logical validity without relying on formal mathematical operations. By applying our math-derived steering vectors to these datasets, we aim to evaluate the transferability of our reasoning-aware intervention across distinctly different reasoning modalities.

Table 5 summarizes the performance of our method on the two non-mathematical reasoning datasets. Across both benchmarks, the application of our math-derived steering vectors yields consistent and stable improvements in reasoning accuracy over the baselines. These results substantiate the hypothesis that our steering vectors do not merely overfit to domain-specific mathematical heuristics or vocabulary. By effectively correcting the fundamental structural errors introduced by reasoning drift, our intervention demonstrates broader applicability to general reasoning tasks beyond the scope of its original extraction data.

## J  CASE STUDY

### J.1  ORIGINAL SOLUTION CASE

Below shows an original solution that leads to a failed outcome.

> (Initial Question) Solve this math problem step by step. You MUST put your final answer in boxed{}. Problem: Steve loves playing video games. His parents get him a console along with 5 games for his birthday. He saves up enough money to buy 1 game per month for a year, and then the following year he starts buying 2 games a month. For the third year he buys 4 games a month as he has a new part-time job that makes him more money. He also gets 5 games for Christmas every year. How many games does Steve have after 3 years? Solution: <think>
> (Plan Generation) To determine how many games Steve has after 3 years, let's break down his purchases and the games he receives each year.
> (Active Computation) **Year 1:**
> - **Monthly Purchases:** 1 game per month × 12 months = 12 games
> - **Total Games in Year 1:** $12 + 5 = 17$ games
> ... ...
> (Active Computation) **Year 2:**
> ... ...
> (Active Computation) **Year 3:**
> ... ...
> (Activate Computation) **Total Games After 3 Years:**
> - **Year 1:** 17 games
> - **Year 2:** 29 games
> - **Year 3:** 53 games
> - **Cumulative Total:** $17 + 29 + 53 = 99$ games.
> (Final Answer Emission) Therefore, Steve has a total of 99 games after 3 years.
> </think>

## J.2 STEERED SOLUTION CASE

Below shows a steered solution that leads to a successful outcome.

---

(Initial Question) Solve this math problem step by step. You MUST put your final answer in boxed{}. Problem: Steve loves playing video games. His parents get him a console along with 5 games for his birthday. He saves up enough money to buy 1 game per month for a year, and then the following year he starts buying 2 games a month. For the third year he buys 4 games a month as he has a new part-time job that makes him more money. He also gets 5 games for Christmas every year. How many games does Steve have after 3 years? Solution: <think>

(Problem Setup) Okay, let me try to figure out how many games Steve has after 3 years.

(Fact Retrieval) Hmm, he gets the console and 5 games for his birthday.

(Plan Generation) First, I need to break down what he does each year.

(Fact Retrieval) In the first year, he buys 1 game per month for a year.

... ...

(Active Computation) So for the first year, he gets 12 games from his monthly purchases and 5 games from Christmas.

... ...

(Uncertainty Management) But hold on, does that include the initial 5 games from the birthday?

(Uncertainty Management) But wait, that doesn't include the initial 5 games from the birthday.

... ...

(Active Computation) So 5 + 17 + 29 + 53. 5 + 17 is 22, 22 + 29 is 51, 51 + 53 is 104. So 104 games.

(Self Checking) But earlier, I thought of 17 + 29 + 53 = 99, which would not include the initial 5. So, perhaps, the correct answer is 104.

... ...

(Final Answer Emission) So, I'm confident that the answer is 104.

</think>

---

