# OpenReview forum: "Characterizing and Mitigating Reasoning Drift in Large Language Models"
_ICLR.cc/2026/Conference — ICLR 2026 Poster_

### Official Review · Reviewer_iJ4q · 2025-10-18

**Soundness:** 3
**Presentation:** 3
**Contribution:** 3
**Rating:** 8
**Confidence:** 3

**Summary:**

This paper first characterizes the failure modes of LLM reasoning patterns, and then proposes to steer the generation with pattern-specific steering vectors.

To characterize the failure patterns, the paper trains a classifier to label each reasoning step into one of the 9 categories (e.g. "initial question". "Problem setup", "Uncertainty management" etc). It identifies model-specific failure modes, e.g. Llama tends to get stuck in self-loops, and Qwen tends to make erractic jumps.
- Specifically, the classifier is a DistillBERT trained on the reasoning trajectories from the Math-Rollout datasets and the category labels therein.

Based on these findings, the paper constructs steering vectors by taking the difference in activations between a desired step and an undesired step. At inference time, the model computes per-step steering vector adaptively and adds the steering vector to the activation.
- The layer to intervene is chosen to be the one that incurs the most change in the activation.
- The adaptive steering vector is a weighted sum of all steering vectors, where the weights are based on the similarity the current activation and the steering vectors.

The experiments are based on R1-distilled Llama and Qwen, using 4 math datasets (i.e. GSM8k, AIME2024, AIME2025, GPQA-Diamond). They show that:
- The steering method effectively changes the transition probabilities among the 9 categories.
- Steering improves math reasoning performance over baselines (vanilla CoT, self-consistency, and PiCSAR).
- The steering method is applicable to Llama and Qwen without distillation, though the improvement is much smaller.

**Strengths:**

- The analysis on characterizing per-step patterns can be useful for diagnosing models, which helps inform intervention which the paper subsequently explores.
- The proposed inference-time intervention is lightweight and can be practically useful. Results on the 4 math datasets are promising: the intervention achieves noticeably better accuracy without requiring resampling / parallel sampling.
  - Notably, the steering vectors computed on Math-Rollout are applicable to 4 datasets which can be considered OOD.
- The writing is easy to follow.

**Weaknesses:**

- Sec 4.4 / Table 2: The effectiveness of steering seems to decrease significantly with non-distilled models.
- The improvement on AIME2025 (i.e. the newest benchmark) is much more moderate than other datasets. This might suggest that the method might be less effective when the applicable is truly OOD -- which is understandable though worth noting.
- The intervention requires white-box access to the activations, which can be strong for certain practical considerations.

**Questions:**

- To clarify, is it correct that: the trained DistilBERT classifier is for annotating rollouts that are not in Math-Rollouts, and the classifier is used only for analysis and not during inference.
- line 297: should it be $S^+$ has the label $i$?
- Sec 4.4 / Table 2: are the steering vectors extracted per-model? What do you think makes the performance gain much weaker now (e.g. is this related to whether the models have gone through distillation)? What are the performance of SC and PiCSAR?
- How to improve the effect of steering in later stage? For example, is this because one token matters less when the attention is put on more tokens compared to at the beginning? Would additionally biasing the attention weights be helpful?

---

> ### Author Response · Authors · 2025-11-25
> **Response to Reviewer iJ4q**
>
> We thank the reviewer for the very careful reading and the constructive suggestions. We are glad that the reviewer finds our method sound. The reviewer also points out several meaningful directions for strengthening the work. Below we detail our responses point by point.
>
> > **Q1. To clarify, is it correct that: the trained DistilBERT classifier is for annotating rollouts that are not in Math-Rollouts, and the classifier is used only for analysis and not during inference.**
>
> **A1.** We thank the reviewer for the question and are happy to clarify the role of the DistilBERT classifier. In our framework, the classifier serves **two purposes**. First, as the reviewer correctly inferred, it is used offline to annotate rollouts that are not covered by the original Math-Rollout dataset, enabling us to assign functional labels and conduct the diagnostic analysis at scale. Second, the classifier is also used **during inference**: since our method intervenes online, we need to know which functional phase the model is currently in so that we can decide whether and how to apply steering. The classifier provides a lightweight way to infer the current step type, avoiding the impractical overhead of repeatedly querying a large LLM for self-labeling at each step.
>
> > **Q2. line 297: should it be S+ has the label i?**
>
> **A2.** Thanks for pointing this out. In our framework, the previous step S_t-1 is assigned a label i, and this label is used to guide the token-level generation in the current step S_t. This is because our method intervenes online, token by token, so we rely on the functional label of the preceding step to inform how the current step’s tokens should be steered. We will revise the text to make this point explicit.
>
> > **Q3. Sec 4.4 / Table 2: are the steering vectors extracted per-model? What do you think makes the performance gain much weaker now (e.g. is this related to whether the models have gone through distillation)? What are the performance of SC and PiCSAR?**
>
> **A3.** We appreciate the opportunity to clarify the issue behind the relatively small gains reported on the base models.
>
> First, the evaluation setting used in Table 2 is intentionally a **strict test of generalization**: instead of re-estimating steering vectors on each target model, we directly reuse the vectors learned on the DeepSeek-R1–distilled models and apply them to the corresponding non-distilled base models. Our motivation is that, once the vectors are released, other researchers may be interested in “plug-and-play” reuse on the same model family without recomputing everything from scratch. Thus, RQ3 is designed to answer the practical question: _what happens under this more demanding, zero-retraining transfer scenario?_ We realize that this motivation was not clearly communicated due to space limitations, and we will make this explicit in the revised version.
>
> Regarding the magnitude of the improvements, several factors need to be emphasized. The steering vectors are learned in a representation space shaped by the R1 distillation process, where the model exhibits an explicit “thinking” mode with structured reasoning traces, whereas the base models do not follow this explicit reasoning format. This **paradigm gap** naturally weakens the effect of steering when vectors are transferred across these spaces. From this perspective, small but consistent positive gains are already informative: they suggest that the vectors are not merely capturing characteristics of a single distilled checkpoint, but encode some lower-level regularities about failure-prone trajectories that partially transfer across related models.
>
> We will expand the discussion in RQ3 to clarify this stricter transfer setting, analyze the expected impact of the gap, and more carefully contextualize the observed effect sizes.
>
> Regarding the reviewer’s question about the performance of SC and PiCSAR, we refer to the results already reported in PiCSAR. For completeness, we will summarize the relevant metrics to Table 2.
>
> > **Q4. How to improve the effect of steering in later stage? For example, is this because one token matters less when the attention is put on more tokens compared to at the beginning? Would additionally biasing the attention weights be helpful?**
>
> **A4.** We thank the reviewer for the insightful suggestion regarding enhancing the steering effect for later-stage tokens. We observe that the weaker impact in later tokens is consistent with the effect illustrated in Figure 1: early changes have a more pronounced influence on the final outcome, while intervening at later sentences is less likely to alter the conclusion. As the reviewer noted, this is likely related to the accumulation of contextual information from previous tokens. We agree that exploring mechanisms such as biasing attention weights to strengthen steering for later-stage tokens is an interesting and promising direction for future research.

---

> ### Comment · Reviewer_iJ4q · 2025-11-26
>
> Thank you for the clarifications!
>
> Since the paper has a large spread of scores, I'd like to justify my rating.
>
> For an empirical paper like this, I care about 1) whether the experiments are reasonably thorough, and 2) whether the paper provides concrete takeaways or algorithms leading to practical improvements. I think this paper achieves both, though it could benefit from clarifications mentioned by all reviewers. I hence vote to accept.
>
> Some notes after reading the rebuttals to other reviewers:
> - The results on 1) random steering vector baselines and 2) non-math reasoning datasets help strengthen the paper.
>
> - Whether focusing on math is too limited: I'm okay with limiting the scope to math reasoning, which generally requires multi-step reasoning, as opposed to "general" reasoning tasks that are mostly based on factual knowledge. Elaborating on this point could strengthen the paper, e.g. by showing that the proposed method provides more gain when the number of reasoning steps (which needs to be properly defined) increases.
>
> - Re out-of-distribution: I think of different math datasets as out-of-distribution -- this is the classic notion of OOD, i.e. same "domain" (both in a math sense and in an English sense), but different distributions within the domain. I think transferring from math to other domains e.g. programming might be 1) too much to ask for, and 2) not practically necessary, as models are seldom required to generalize to an unseen domain.

---

### Official Review · Reviewer_iuYD · 2025-10-30

**Soundness:** 2
**Presentation:** 2
**Contribution:** 2
**Rating:** 4
**Confidence:** 3

**Summary:**

This paper introduces "Reasoning Drift," a failure mode in LLM chain-of-thought processing where models adopt flawed functional patterns. To address this, the authors propose Reasoning-Aware Activation Steering (RAAS), an inference-time method that uses a pre-computed library of steering vectors to guide model activations away from pathological states. The method is evaluated on several mathematical reasoning benchmarks and demonstrates significant accuracy improvements.

**Strengths:**

1) The paper provides a rigorous, data-driven characterization of "Reasoning Drift." The analysis of functional state transitions offers a valuable new lens for understanding LLM reasoning failures that moves beyond surface-level error analysis.
2) The proposed RAAS intervention is elegantly designed and directly motivated by the preceding diagnostic analysis. As a lightweight, interpretable, inference-time technique, it presents a practical alternative to computationally expensive methods like large-scale sampling.
3) The method achieves consistent and substantial accuracy gains across multiple challenging mathematical reasoning benchmarks, including complex, out-of-distribution datasets like AIME.

**Weaknesses:**

1) The core methodology relies on a strong and unproven assumption. The diagnostic analysis and steering vectors are derived from sentence-level representations (using mean-pooled activations; see Eq. 2), yet the intervention is applied at the token level (see Eq. 4, 6). The paper provides no evidence to validate that a single token's activation can be meaningfully compared to the averaged activation of an entire sentence, calling the intervention's foundational principle into question.
2) The paper demonstrates a correlation between the RAAS intervention and improved performance but fails to establish causation. The accuracy gains could be a byproduct of an uncontrolled factor, such as a regularization effect from the activation shifts, rather than the specific correction of functional transitions as claimed. The lack of critical control experiments (e.g., applying randomly selected steering vectors) leaves the mechanistic claims unverified. Maybe adopting significance tests (such as bootstrap) can make results more convincing.
3) The paper's claim of generalization to "out-of-distribution tasks" may be misleading. The steering vectors are extracted from a mathematics dataset (Math-Rollout), and all evaluation benchmarks (GSM8K, AIME, GPQA) are also within the mathematical reasoning domain (see Sec 4.1). This constitutes within-domain generalization at best, and there is no evidence to suggest the framework would be effective in other domains such as programming.

**Questions:**

The questions refer to the weaknesses.

---

> ### Author Response · Authors · 2025-11-25
> **Response to Reviewer iuYD (Part 1)**
>
> We sincerely thank the reviewer for the thorough review and insightful suggestions. Below we respond to each concern and clarify the points in detail.
>
> > **Q1. The core methodology relies on a strong and unproven assumption. The paper provides no evidence to validate that a single token's activation can be meaningfully compared to the averaged activation of an entire sentence, calling the intervention's foundational principle into question.**
>
> **A1.** This is indeed an important modeling choice, and we agree it deserves a clearer justification. Our design is guided by two considerations:
>
> First, for extracting functional prototypes and steering directions, we follow prior work [1] in representing each reasoning step at the sentence level via mean-pooled activations. This aggregation is intended to capture the **overall semantic** and functional tendency of a step, rather than the potentially noisy or ambiguous signal from any single token. Analogous to classic observations in contextualized embeddings, individual token embeddings can be locally ambiguous, while their pooled or contextualized representations better reflect the role of the entire span in the reasoning process.
>
> Second, during inference, we adopt the token-level steering paradigm that has been widely used in activation steering work [1,2,3]. Since generation unfolds token by token, we need a mechanism that can react online and nudge the model incrementally. In this setting, the steering vector is not meant to pinpoint a single “optimal” token, but rather to act as a coarse directional bias that pulls the token-level activations toward a region of the representation space associated with more desirable functional behavior. Prior studies have shown that such direction-based interventions at the token level can consistently influence higher-level attributes and behaviors, even when the steering directions are estimated at a coarser granularity.
>
> We will clarify this assumption in the revised paper and add an analysis to empirically support this alignment.
>
> > [1] Understanding reasoning in thinking language models via steering vectors. Venhoff, C., Arcuschin, I., Torr, P., Conmy, A., & Nanda, N. arXiv preprint, 2025.
>
> > [2] Semantics-Adaptive Activation Intervention for LLMs via Dynamic Steering Vectors. Wang, W., Yang, J., & Peng, W. International Conference on Learning Representations (ICLR), 2025.
>
> > [3] Steering llama 2 via contrastive activation addition. Panickssery, N., Gabrieli, N., Schulz, J., Tong, M., Hubinger, E., & Turner, A. M. ACL, 2024.
>
> > **Q2. The lack of critical control experiments (e.g., applying randomly selected steering vectors) leaves the mechanistic claims unverified. Maybe adopting significance tests (such as bootstrap) can make results more convincing.**
>
> **A2.** We thank the reviewer for highlighting the importance of control experiments to verify that the gains are not merely due to generic activation perturbations. Following this suggestion, we conducted an additional ablation study using **random steering vectors**. Concretely, we designed three control variants: (1) randomly sampling the source transition type i while keeping the target type j fixed, (2) randomly sampling the target transition type j while keeping the source i fixed, and (3) randomly sampling both the source and target types i and j. The results are summarized in Tab B.
>
> **Tab B. Ablation study of random steering vectors on DS-Llama-8B.**
>
> | | **AIME 2024** | **AIME 2025** | **GPQA-D** |
> | :--: | :--: | :--: | :--: |
> | Vanilla | 34.99 | 25.56 | 50.50 |
> | Ours | 55.56 | 31.70 | 55.52 |
> | Ours + random source i | 50.00 | 30.00 | 51.51 |
> | Ours + random target j | 53.33 | 26.67 | 53.03 |
> | Ours + random source i + random target j | 30.00 | 26.67 | 49.49 |
>
> The new experiments show that partially randomizing a single side (only i or only j) leads to mild degradation, while fully randomizing both directions degrades performance more substantially.
>
> Consistent with this observation, Appendix D and Figure 6 in the paper also provide indirect evidence in the same direction. When the classifier used to assign functional types is weakened (e.g., trained for fewer epochs and thus behaving closer to a noisy or partially random type predictor), the downstream performance degrades accordingly. This further supports that the effectiveness of our method depends on meaningful, accurately identified transition types, rather than on arbitrary perturbations to the hidden states.

---

> > ### Author Response · Authors · 2025-11-25
> > **Response to Reviewer iuYD (Part 2)**
> >
> > > **Q3. The paper's claim of generalization to "out-of-distribution tasks" may be misleading. This constitutes within-domain generalization at best, and there is no evidence to suggest the framework would be effective in other domains such as programming.**
> >
> > **A3.** We agree that our use of the term “out-of-distribution tasks” may be misleading. In the revised version, we will adopt more precise wording and explicitly describe these benchmarks as _out-of-distribution within the math domain_ (e.g., different datasets, difficulty regimes, and formats from Math-Rollout), rather than as fully cross-domain OOD tasks.
> >
> > In addition, to partially probe transfer beyond mathematics, we have added experiments on two non-math reasoning benchmarks, ReClor and LogiQA, which are standard datasets for reading-comprehension-style logical reasoning. Results are shown in Tab C.
> >
> > **Tab C. Comparison on non-math reasoning benchmarks.**
> >
> > | | **ReClor** | **LogiQA** |
> > | :--: | :--: | :--: |
> > | DS-Llama-8B + Vanilla | 46.39 | 58.80 |
> > | DS-Llama-8B + Ours | 48.85 | 61.00 |
> > | DS-Qwen-14B + Vanilla | 65.97 | 78.80 |
> > | DS-Qwen-14B + Ours | 66.51 | 81.40 |
> >
> > We hope that our clarifications and additional analyses adequately address the reviewer’s concerns, and we again appreciate the time and effort invested in evaluating our work.

---

### Official Review · Reviewer_hd5F · 2025-10-31

**Soundness:** 2
**Presentation:** 1
**Contribution:** 2
**Rating:** 2
**Confidence:** 4

**Summary:**

The paper identifies the issue of "Reasoning Drift" (when LLMs use unsuccessful reasoning patterns) and propose Reasoning-Aware Activation Steering (RAAS), an approach based on steering vectors to mitigate this issue. They claim that current methods: 1. require lots of model inferences to implement some kind of best-of-N like mechanism, and 2. don't use LLM internals so we don't understand where they go wrong (although this method doesn't help too much with this). The analysis in the paper is done using the Math-Rollout dataset, a dataset of CoT reasoning traces of a DeepSeek R1 distilled Qwen model on common math datasets, where reasoning sentences are labeled to one of 8 different reasoning categories, e.g., problem setup or plan generation, and also sentences are swapped out in various parts of the CoT to study counterfactual generations. The authors use this dataset to train a DistillBERT model to classify the reasoning category of sentences, and then use this to determine which reasoning category transitions on a sentence by sentence level are useful and not useful, e.g., perhaps self checking after problem setup is not very useful. The RAAS algorithm uses these transitions to design steering vectors and then adds a weighted average of these vectors when the reasoning category transition is expected to not be useful. Unrelatedly, they identify a "funneling effect", where on this math reasoning dataset, changing sentences early in the CoT have more influence over final answers than later sentences.

**Strengths:**

- The analysis of the Math-Rollout dataset to identify the reasoning drift is conceptually interesting. The "funneling effect" and the distinction between model-specific failure signatures (Inertial vs. Chaotic Drift) are also interesting.

- Use of activation steering to the process of reasoning (RAAS) at test-time is a creative idea. This makes the method computationally efficient.

**Weaknesses:**

- The most significant weakness is the paper's heavy reliance on the Math-Rollout dataset and limited evaluation. This dataset is specific to the math domain and derived from only two specific distilled models (R1-Distill-Llama/Qwen). It is unclear if the identified drift patterns, the 8-category taxonomy, or the derived steering vectors generalize beyond this domain and beyond these models. The gains on non-distilled models are too small to justify generalizability.

- The paper suffers from significant clarity issues, making it difficult to follow, and the writing style often lacks precision and lots of loose terminology. Key terms such as "universal functional tendencies" and "functional shifts" are vague and not formally defined. The introduction frames the problem confusingly. For example, it describes how changing sentences in a CoT can lead to "entirely different outcomes" as if this is inherently problematic, whereas this is obvious and expected in a reasoning process. The specific "negative impact of stochasticity" they aim to address is not clearly articulated. The figures are confusing and poorly presented.

- Experimental details for the DistilBERT classifier, a critical component of RAAS, are largely missing from the main paper and sparse even in Appendix D.

**Questions:**

- How dependent are the identified functional transitions and drift patterns on the specific characteristics of the Math-Rollout dataset and the DeepSeek distillation process? Can it extend to other model families and tasks?

- In Figure 3, what do the neon blue and purple outlines around certain rectangles signify? The caption is incomplete and the text does not clarify this.

- In the introduction, the authors seem to treat the fact that changing sentences in a CoT leads to different outcomes as a problem stemming from stochasticity. Can the authors clarify why this is viewed as a "negative impact"? Isn't it expected that different reasoning steps lead to different conclusions?

- Given the very small performance gains on the base models (Table 2), does this suggest that the steering vectors are primarily capturing behaviors specific to the distillation process or the specific dataset, rather than generalizable reasoning principles?

- Writing needs massive improvement. There is a lot of jargon phrases like 'reasoning drift', 'funneling effect', 'universal functional tendencies' and 'functional shifts' to name a few. They don't really mean much and are never formally defined. Also the intro seems to use a lot of such repetitive language but doesn't convey the main results. Also, RAAS uses Reasoning-Aware and Reasoning-Adaptive in different parts.

---

> ### Author Response · Authors · 2025-11-25
> **Response to Reviewer hd5F (Part 1)**
>
> We appreciate the reviewer for the very careful reading and the constructive suggestions. Below we detail our responses point by point.
>
> > **Q1. The most significant weakness is the paper's heavy reliance on the Math-Rollout dataset and limited evaluation.**
>
> > **How dependent are the identified functional transitions and drift patterns on the specific characteristics of the Math-Rollout dataset and the DeepSeek distillation process? Can it extend to other model families and tasks?**
>
> **A1.** Regarding the concern about our reliance on the Math-Rollout dataset and the two distilled models, we agree that our study is primarily grounded in this setting, and this choice is deliberate rather than accidental.
>
> First, research on LLM reasoning is still largely “black-box”, and there are very few open datasets that expose fine-grained trajectories suitable for systematic analysis of failure. Math-Rollout is, to our knowledge, one of the few recent, publicly available resources that provide sufficiently rich step-level rollouts to support this type of investigation.
>
> Second, mathematical reasoning offers a relatively well-structured and interpretable testbed. It is an important and representative subdomain of reasoning where the target outcomes are deterministic, and the solution format is comparatively standardized. **We view math reasoning as a concrete entry point for probing more abstract or general aspects of reasoning**, with the goal of uncovering patterns and tools that can inspire future work, rather than claiming to solve general reasoning in full generality.
>
> On top of this focused setup, we have made conscious efforts to probe generalization beyond the exact training conditions. Specifically, the steering vectors are learned from Math-Rollout but evaluated on different math benchmarks that can be viewed as out-of-distribution (GSM8K, AIME 2024/2025, and GPQA), as well as on the corresponding non-distilled base models.
>
> Furthermore, to explore transfer beyond mathematics, we have added experiments on two non-math logical reasoning datasets, ReClor and LogiQA (see Tab A), which are standard benchmarks for reading-comprehension-style logical reasoning. While the gains on these datasets are modest, they are consistently positive and indicate that our method captures some reusable signals about unproductive trajectories across tasks.
>
> **Tab A. Comparison on non-math reasoning benchmarks.**
>
> | | **ReClor** | **LogiQA** |
> | :--: | :--: | :--: |
> | DS-Llama-8B + Vanilla | 46.39 | 58.80 |
> | DS-Llama-8B + Ours | 48.85 | 61.00 |
> | DS-Qwen-14B + Vanilla | 65.97 | 78.80 |
> | DS-Qwen-14B + Ours | 66.51 | 81.40 |
>
> > **Q2. The paper suffers from significant clarity issues, making it difficult to follow, and the writing style often lacks precision and lots of loose terminology.**
>
> > **Writing needs massive improvement.**
>
> **A2.** Regarding the concern about terminology (e.g., “reasoning drift”, “funneling effect”, “universal functional tendencies”, “functional shifts”), our intention was to study a largely unexplored aspect of model behavior, and these terms were introduced to name phenomena that we empirically observed rather than to reuse existing definitions. In Section 2.3, we provide formal definitions for the step types S_t, rollout types T_t, and the two transition scenarios between S_t-1 → S_t and S_t-1 → T_t.
>
> We agree that the current exposition can be made clearer and less jargon-heavy. In the revised version, we will (i) streamline the terminology, (ii) move the formal definitions earlier and make them more prominent, and (iii) explicitly connect each new term to its mathematical definition and empirical manifestation, so that readers can follow these new concepts without confusion.
>
> > **Q3. In Figure 3, what do the neon blue and purple outlines around certain rectangles signify? The caption is incomplete and the text does not clarify this.**
>
> **A3.** We thank the reviewer for pointing out the confusion around Figure 3. The neon blue and purple outlines are intended to highlight, for each row, the minimum and maximum values, respectively, so that readers can more easily identify the most and least prominent transition patterns in a visually dense matrix. Due to space constraints and an oversight on our side, this detail was not explicitly documented in the caption, which understandably makes the figure hard to interpret. In the revised version, we will update the caption and figure legend to clearly explain the meaning of these outlines and ensure that all visual cues in the figure are explicitly described to avoid further confusion.

---

> ### Author Response · Authors · 2025-11-25
> **Response to Reviewer hd5F (Part 2)**
>
> > **Q4. In the introduction, the authors seem to treat the fact that changing sentences in a CoT leads to different outcomes as a problem stemming from stochasticity. Can the authors clarify why this is viewed as a "negative impact"? Isn't it expected that different reasoning steps lead to different conclusions?**
>
> **A4.** Our intention is not to claim that “different reasoning steps should never lead to different conclusions” in general. Rather, we are concerned with a specific phenomenon observed in deterministic reasoning tasks (e.g., math problems with a single correct answer): when sampling with a non-trivial temperature (e.g., 0.6, as recommended in some model usage guidelines), small changes in the CoT or re-sampling the CoT can lead to semantically different reasoning trajectories that frequently end in different final answers, as also discussed in prior work [1,2]. In such settings, the target solution is fixed, and conceptually different valid reasoning paths should still converge to the same correct outcome.
>
> From this perspective, **strong output variance induced by stochastic sampling reflects a reliability issue**. The model’s behavior is overly sensitive to variations in the generated reasoning, making it difficult to trust its answers in high-stakes or evaluation scenarios. Moreover, **this semantic volatility limits interpretability**. It is unclear which reasoning paths are truly robust (consistently leading to correct answers) and which ones systematically drift toward failure, and existing approaches mostly rely on brute-force sampling and voting or ranking over many CoTs. Our work explicitly focuses on understanding and controlling this aspect of stochasticity. By analyzing rollout trajectories, characterizing drift patterns, and using steering to counteract unproductive trajectories, we aim to make the model’s reasoning both more stable and more interpretable.
>
> > [1] Forking Paths in Neural Text Generation. E Bigelow, A Holtzman, H Tanaka, T Ullman. International Conference on Learning Representations (ICLR), 2024.
>
> > [2] Thought anchors: Which llm reasoning steps matter? Bogdan, P. C., Macar, U., Nanda, N., & Conmy, A. arXiv, 2025.
>
> > **Q5. Given the very small performance gains on the base models (Table 2), does this suggest that the steering vectors are primarily capturing behaviors specific to the distillation process or the specific dataset, rather than generalizable reasoning principles?**
>
> **A5.** We appreciate the opportunity to clarify a potential misunderstanding behind the relatively small gains reported on the base models.
>
> First, the evaluation setting used in Table 2 is intentionally a **strict test of generalization**: instead of re-estimating steering vectors on each target model, we directly reuse the vectors learned on the DeepSeek-R1–distilled models and apply them to the corresponding non-distilled base models. Our motivation is that, once the vectors are released, other researchers may be interested in “plug-and-play” reuse on the same model family without recomputing everything from scratch. Thus, RQ3 is designed to answer the practical question: _what happens under this more demanding, zero-retraining transfer scenario?_ We realize that this motivation was not clearly communicated due to space limitations, and we will make this explicit in the revised version.
>
> Regarding the magnitude of the improvements, several factors need to be emphasized. The steering vectors are learned in a representation space shaped by the R1 distillation process, where the model exhibits an explicit “thinking” mode with structured reasoning traces, whereas the base models do not follow this explicit reasoning format. This **paradigm gap** naturally weakens the effect of steering when vectors are transferred across these spaces. From this perspective, small but consistent positive gains are already informative: they suggest that the vectors are not merely capturing characteristics of a single distilled checkpoint, but encode some lower-level regularities about failure-prone trajectories that partially transfer across related models.
>
> We will expand the discussion in RQ3 to clarify this stricter transfer setting, analyze the expected impact of the gap, and more carefully contextualize the observed effect sizes.

---

> > ### Author Response · Authors · 2025-11-25
> > **Response to Reviewer hd5F (Part 3)**
> >
> > > **Q6. Experimental details for the DistilBERT classifier, a critical component of RAAS, are largely missing from the main paper and sparse even in Appendix D.**
> >
> > **A6.** Regarding the DistilBERT classifier, we respectfully think the current level of detail is adequate for supporting its use in our pipeline. First, Table 3 and Appendix C already summarize the dataset characteristics (e.g., sample counts and class definitions), while Table 4 and Appendix D describe the training configuration and outcomes of the classifier, including split ratios, epochs, learning rate, and standard metrics such as precision, recall, F1-score, and macro/weighted averages. Given that DistilBERT is a widely adopted and mature architecture, we believe this level of reporting is sufficient and allows the main paper to focus on the core contributions that are less standardized and rely on more novel design choices.
> >
> > Second, Table 6 further analyzes how classifiers with different training epochs (thus different classification capacities) affect downstream reasoning. This sensitivity analysis suggests that the 3-epoch trained classifier provides stable and adequate performance for our method.
> >
> > We hope that our clarifications and additional analyses adequately address the reviewer’s concerns, and we again appreciate the time and effort invested in evaluating our work.

---

### Official Review · Reviewer_kY9T · 2025-11-03

**Soundness:** 3
**Presentation:** 4
**Contribution:** 3
**Rating:** 6
**Confidence:** 4

**Summary:**

This paper exploits a rich dataset of math reasoning rollouts to trace reasoinng trajectories that cause failure or success. Through defining corrective and erroneous rollouts, it identifies the promising and unpromising steps and studies the transitions of the model between step categories. This gives insight into which transitions are useful and which ones might hurt performance. Interestingly, this reveals different failure patterns in the Qwen and Llama models. Moreover, the paper proposes a method for detecting the undesired drift in reasoning and learns steering vectors that prove to be effective to correct reasoning trajectories in inference time.

**Strengths:**

The paper provides a granular study of the frequent reasoning patterns, which is helpful in studying the failure modes of reasoning in large language models. Through using a rich dataset, it is able to identify model-specific patterns. Then, it extends its study to a practical method that detect drifts in reasoning and corrects them at inference time with interventions with steering vectors. Overall, the work both brings clarity to understanding reasoning behavior of language models and provides practical solutions.

**Weaknesses:**

The proposed methods, while being effective in experiments, do not seem very well-motivated. It is not clear why defining the DriftScore in the way it is defined is plausible. Moreover, it seems that v_{i->j} can be computed from a_{i, j}^+ and a_{i, j}^-. Finally, the application of the method to other methods RQ3 could be explored more thoroughly to understand the gap between the results.

**Questions:**

1. Could you please elaborate on why the improvements for other models (RQ3) is less significant? Are the transition patterns different?
2. Could you explain the reason behind the definition of DriftScore? Is it just a heuristic or it is well-motivated?

---

> ### Author Response · Authors · 2025-11-25
> **Response to Reviewer kY9T**
>
> We sincerely thank the reviewer for the thorough review and insightful suggestions. Below we respond to each concern and clarify the points in detail.
>
> > **Q1. Could you please elaborate on why the improvements for other models (RQ3) is less significant? Are the transition patterns different?**
>
> **A1.** We appreciate the opportunity to clarify a potential misunderstanding behind the relatively small gains reported on the base models.
>
> First, the evaluation setting used in Table 2 is intentionally a **strict test of generalization**: instead of re-estimating steering vectors on each target model, we directly reuse the vectors learned on the DeepSeek-R1–distilled models and apply them to the corresponding non-distilled base models. Our motivation is that, once the vectors are released, other researchers may be interested in “plug-and-play” reuse on the same model family without recomputing everything from scratch. Thus, RQ3 is designed to answer the practical question: _what happens under this more demanding, zero-retraining transfer scenario?_ We realize that this motivation was not clearly communicated due to space limitations, and we will make this explicit in the revised version.
>
> Regarding the magnitude of the improvements, several factors need to be emphasized. The steering vectors are learned in a representation space shaped by the R1 distillation process, where the model exhibits an explicit “thinking” mode with structured reasoning traces, whereas the base models do not follow this explicit reasoning format. This **paradigm gap** naturally weakens the effect of steering when vectors are transferred across these spaces. From this perspective, small but consistent positive gains are already informative: they suggest that the vectors are not merely capturing characteristics of a single distilled checkpoint, but encode some lower-level regularities about failure-prone trajectories that partially transfer across related models.
>
> We will expand the discussion in RQ3 to clarify this stricter transfer setting, analyze the expected impact of the gap, and more carefully contextualize the observed effect sizes.
>
> > **Q2. Could you explain the reason behind the definition of DriftScore? Is it just a heuristic or it is well-motivated?**
>
> **A2.** Thanks for the comment, and we clarify why we define DriftScore in its current form.
>
> First, from the rollout analysis we know that different transition types have systematically different downstream success rates. This gives us a set of offline “drift patterns”. However, at inference time we cannot wait until the entire next step S_t is finished and then re-classify it; **we must decide on the fly how to guide the model while it is still generating**. The previous step type S_t-1 is easy to obtain by classifying already generated text, but the future type St is not directly observable. **DriftScore is therefore introduced as an online surrogate that estimates the current tendency of the hidden state toward different step types**, by measuring its similarity (via cosine similarity in our implementation) to the learned drift patterns. We only intervene when the current representation is close to historically failure-prone patterns, which is consistent with our description in lines 316–323.
>
> Second, the “bad” patterns are not absolutely bad in all contexts. For example, a small number of computation→computation transitions is often a necessary part of correct reasoning, while excessive repetition of such transitions is strongly associated with failure. For this reason, DriftScore is defined as a continuous scalar coefficient (see Eq. 5) rather than a binary trigger. It controls the strength of the steering intervention. If the current state only mildly resembles drift patterns, the score (and thus the intervention) is small; if it closely matches high-risk patterns, the score becomes larger; and when it is closer to good prototypes, the score is exactly zero and no intervention is applied.
>
> In summary, DriftScore is not an arbitrary heuristic but a statistically motivated and practically necessary surrogate that (i) projects offline transition statistics into an online, single-pass drift indicator, and (ii) enables graded, context-sensitive control over the steering vectors instead of bluntly suppressing entire step types.
>
> We hope that our clarifications and additional analyses adequately address the reviewer’s concerns, and we again appreciate the time and effort invested in evaluating our work.

---

### Author Response · Authors · 2025-12-02
**Note to Area Chairs**

Dear AC,

Thank you for handling our submission. Below is a summary of the key points raised by the four reviewers and how our rebuttal addresses them.

Across reviews, the strengths highlighted include:

* A valuable new lens for understanding LLM reasoning that moves beyond surface-level error analysis (kY9T, iuYD).
* A rigorous and data-driven characterization of per-step reasoning patterns and functional state transitions (kY9T, hd5F, iuYD, iJ4q).
* A practical analysis-to-intervention solution, where RAAS is directly motivated by the diagnostic study (kY9T, iuYD, iJ4q).
* A lightweight, interpretable inference-time method that brings consistent and substantial performance gains (hd5F, iuYD, iJ4q).

The main concerns focused on (1) weaker improvements on non-distilled models, (2) generalization scope beyond math, (3) the lack of random-vector controls, and (4) the sentence-level vs. token-level representation assumption. In our rebuttal, we (1) provided analysis explaining the gap between distilled and non-distilled models, (2) expanded experiments to non-math reasoning datasets (ReClor, LogiQA), (3) added three random-steering baselines showing clear degradation, and (4) clarified the design rationale for sentence-level representations. We also clarified the role of the DistilBERT classifier and corrected minor presentation issues noted by the reviewers.

We note that Reviewer iJ4q confirmed that the experiments are reasonably thorough, that the work provides concrete takeaways, and that the added baselines and non-math results strengthen the work. They further noted that focusing on multi-step mathematical reasoning is a reasonable scope, and that OOD within the math domain is the appropriate notion for this problem setting.

We hope this summary helps contextualize our responses and clarifications.

Best Regards,

The authors

---

### Meta-Review · Area_Chair_eacc · 2026-01-06

**Summary:**

This paper presents a detailed study on reasoning drift in LLMs and proposes a novel intervention method, RAAS, to mitigate it. Reviewers highlighted the work's rigorous, data-driven analysis of failure patterns and its practical, lightweight inference-time solution, which demonstrates substantial gains on mathematical reasoning benchmarks. Key concerns centered on the scope of evaluation, the transferability to non-distilled models, and the foundational assumptions linking sentence-level and token-level representations. In their rebuttal, the authors provided additional experiments on non-mathematical reasoning tasks and random-vector controls, while offering clarifications on the methodological design and the interpretation of results. The discussion reveals that while the work is grounded in a specific domain, the analysis is insightful and the proposed method is effective and well-motivated. After considering the rebuttal and the engaged discussion among reviewers, the consensus is that the paper's contributions merit acceptance.

**Reviewer Concerns:**

The authors' rebuttal has substantively engaged with several key concerns raised during the review. Critiques regarding the lack of control experiments and the narrowness of evaluation have been directly addressed through the introduction of new baselines with random steering vectors and the inclusion of results on non-mathematical reasoning datasets like ReClor and LogiQA. Clarifications on the rationale behind the DriftScore definition and the sentence-level versus token-level representation assumption have also strengthened the methodological exposition.

However, some underlying reservations remain only partially mitigated. The core premise of transferring insights from distilled to base models, while pragmatically motivated, still results in notably weaker performance gains, highlighting a fundamental limitation in generalizability. Furthermore, while the terminology has been clarified, the foundational claim that the identified "drift" patterns capture a universal reasoning flaw, rather than a dataset-specific artifact, relies heavily on the chosen taxonomy and the proprietary Math-Rollout data. The work provides a compelling proof-of-concept within its scope, but these aspects suggest its broader implications require further validation.

**Reviewer Scores:**

| Reviewer | Initial Score | Predicted New Score | Reasoning |
|-|-|-|-|
| kY9T | 6 | 7 | The authors' clarification on the DriftScore definition and the stricter generalization setting for non-distilled models addressed the core methodological concerns, likely increasing confidence in the work's soundness. |
| hd5F | 2 | 3 | While new non-math experiments and writing improvements were added, the heavy dependence on the Math-Rollout dataset and limited gains on base models still pose significant limitations, preventing a major score shift. |
| iuYD | 4 | 6 | The addition of random-steering baselines and expanded evaluation to logical reasoning tasks directly countered key weaknesses regarding controls and scope, elevating the paper above the acceptance threshold. |
| iJ4q | 8 | 8 | Already strongly supportive, the reviewer's endorsement was reinforced by the rebuttal's new analyses, seeing the paper as a clear accept with concrete contributions. |

---

### Decision · Program_Chairs · 2026-01-26

Accept (Poster)